# Psychological Implications to the Therapy of Systemic Lupus Erythematosus

**DOI:** 10.3390/ijerph192316021

**Published:** 2022-11-30

**Authors:** Katarzyna Warchoł-Biedermann, Ewa Mojs, Dorota Sikorska, Przemysław Kotyla, Grażyna Teusz, Włodzimierz Samborski

**Affiliations:** 1Department of Clinical Psychology, Poznan University of Medical Sciences, 60-812 Poznan, Poland; 2Department of Rheumatology, Rehabilitation and Internal Diseases, Poznan University of Medical Sciences, 61-545 Poznan, Poland; 3Department of Internal Medicine, Rheumatology and Clinical Immunology, Medical University of Silesia Katowice, 40-635 Katowice, Poland; 4Faculty of Educational Studies, Adam Mickiewicz University, 61-712 Poznan, Poland

**Keywords:** medication adherence, surveys and questionnaires, lupus nephritis, rheumatology

## Abstract

Systemic lupus erythematosus (SLE) is a chronic and multi-systemic autoimmune disease, which has a deleterious impact on patients’ psychological well-being. This paper aims to review the existing literature on empirical research on psychological outcomes of SLE and psychological interventions to improve well-being in SLE patients. A search of significant English language articles was conducted in PubMed, Medline, ScienceDirect, Scopus, and ResearchGate databases. Titles and abstracts were screened for the relevant terms, including “systemic lupus erythematosus”, “childhood-onset systemic lupus erythematosus”, “juvenile systemic lupus erythematosus”, “lupus nephritis”, and their respective synonyms along with “depression”, “anxiety”, “fatigue”, “medical adherence”, “health-related quality of life”, “self-management” or “intervention”. The articles were evaluated by independent reviewers and the lists of eligible publications were compared whilst disagreements were settled by discussion. Of the 59 publications sought for retrieval, 35 papers were shortlisted based on predefined inclusion/exclusion criteria. They were classified according to their content and the methodology applied. Research topics including “anxiety and depression in SLE” and “self-management interventions for SLE patients” were identified and are presented in this review. As the prognosis and life expectancy of SLE patients are improving, further research on the psychological outcomes of SLE and the evidence-based psychological interventions to improve patients’ well-being are justified.

## 1. Introduction

Systemic lupus erythematosus (SLE) is a connective tissue disease of unknown origin and multi-organ clinical presentation and unpredictable course [1]. SLE is a disease distributed worldwide, which occurs in both genders, and all racial/ethnic and age groups. However, higher rates are observed in adults, females, and non-Caucasians [2]. Genetic, environmental, and sociodemographic factors are responsible for the variable course and outcome of the disease [3]. SLE includes many symptoms, such as cutaneous, haematological, and renal disease. Clinical heterogeneity and unpredictable course and flares are characteristics of SLE [1,2,3,4,5,6].

One of the major problems in SLE patients is neuropsychiatric disease, which has one of the heaviest weights in the measurement of SLE disease activity [7]. The prevalence of neuropsychiatric systemic lupus erythematosus (NPSLE) is estimated to be between 37 and 95% [8]. NPSLE refers to a series of neurological and psychiatric symptoms directly related to SLE. Neuropsychiatric disease includes heterogeneous manifestations involving both the central and peripheral nervous systems [7]. Although in 1999 the American College of Rheumatology (ACR) proposed a set of definitions for the neuropsychiatric involvement in SLE with the intention of homogenizing the terminology, the unequivocal diagnosis remains a challenge. This is due to the multiple neuropsychiatric disease presentations and the severity of symptoms [8].

Additionally, it seems that it is not only patients who meet the criteria for the diagnosis of NPSLE who present with a whole range of psychological symptoms [7]. Even lupus patients in remission often report residual symptoms, such as chronic pain (not only organic) and fatigue [9,10]. Similarly, even successfully treated SLE patients are also at a high risk for depression and anxiety [11]. A clinically significant problem is also decreased sleep quality in this group of patients [12]. All of these symptoms are difficult to clearly diagnose and rank in order but can significantly affect the quality of life of patients with SLE [13].

Therefore, psychological symptoms represent a challenge for the treating physician in terms of diagnosis and treatment. Targeted symptomatic therapy is indicated according to the type of neuropsychiatric manifestations (e.g., antipsychotics for psychosis and anxiolytics for anxiety disorder) [14,15]. However, comprehensive care requires not only pharmacological therapy. Therefore, the role of the psychologist, as a therapist who coordinates psychological care, also appears important [16,17]. The goal of this paper is to review the current empirical research on the psychological outcomes of SLE and the efficacy of psychological interventions to improve well-being in SLE patients.

## 2. Methods

A scholarly literature search to identify English language peer-reviewed original publications on the psychological outcomes of SLE and the psychological interventions to improve well-being in SLE patients released between January 2007 and July 2022 was conducted across PubMed, Medline, ScienceDirect, Scopus, and ResearchGate databases. The review methodology was formally discussed and agreed upon within the research group. This review aimed to collect the evidence on the psychological impact of SLE and the efficacy of psychological interventions in SLE patients. Consequently, the goal to examine the psychological effects of lupus and the results of psychological interventions for lupus patients from the point of the diagnostic disclosure and then across the treatment trajectory was carefully addressed during the selection process. The search terms included both standardized medical subject headings and keywords, such as “systemic lupus erythematosus”, “SLE”, “childhood-onset systemic lupus erythematosus”, “cSLE”, “juvenile systemic lupus erythematosus”, “JSLE”, “lupus nephritis”, and “LN”, along with “depression”, “anxiety”, “fatigue”, “medical adherence”, “health-related quality of life”, “patient”, “self-management” or “intervention”, “psychotherapy”, “therapy”, “treatment”, “psychology”, cause”, ”factor”, “determinant”, and “variable”. The terms were intermingled with Boolean operators and the search queries were combined so that several potential search strategies could be developed to find the relevant results. Papers on the psychological outcomes of SLE were included if they presented prospective, retrospective, longitudinal, observational, or cross-sectional studies. Studies that were descriptive based on qualitative assessments or small sample size investigations without power of the test calculations were also included. Publications on the psychological interventions for lupus patients were included if they set forth single- or double-blinded and non-blinded randomized controlled trials. In turn, investigations exploring the psychological outcomes or the psychological interventions for autoimmune and rheumatologic diseases or autoimmune/rheumatologic disease patients involving a lupus subgroup were included if they highlighted the results for this subgroup. Meta-analyses, reviews, systematic reviews, scoping reviews, and “grey literature”, such as conference reports, commentaries, theses, and non-peer-reviewed publications were excluded from the review. In order to establish a more conclusive evidence of contemporary literature, citation information and published research profiles of key authors were screened for relevant publications. Three independently working researchers reviewed the titles and abstracts yielded by this comprehensive search and selected articles in accordance with the predefined inclusion and exclusion criteria. Subsequently, 65 potentially relevant articles were retrieved for further analysis. The unmasked lists of eligible publications were compared and any disagreement between reviewers was settled through a consensus discussion with a fourth reviewer. The reviewers assessed the full text of the articles retained after the title/abstract screening. Of the 59 articles initially identified through the database search and sought for retrieval, 35 papers that addressed the subject of the review and fulfilled the criteria of methodologically robust research were selected. The details of the screening process, including reasons for full-text exclusion, are delineated in Figure 1.

## 3. Results

The initial search yielded 65 articles after removing duplicate records and articles deemed ineligible. Fifty-nine of these articles were sought for retrieval whilst only 35 articles that complied with the inclusion criteria were included. Twenty-six studies focusing on the psychological outcomes of lupus involved 7302 subjects with lupus [18,19,20,21,22,23,24,25,26,27,28,29,30,31,32,33,34,35,36,37,38,39,40,41,42,43] while nine interventional studies involved 431 subjects with SLE [44,45,46,47,48,49,50,51,52]. Because of a diverse nature of retrieved investigations and the heterogeneity of data in terms of the type of intervention technique, research procedures, and outcome evaluation measures, a meta-analysis could not be performed. The papers were classified on the basis of their contents and methodology. The following research topics were identified and are presented in this review: (1) “Anxiety, depression and sleep quality”; (2) “Fatigue and pain”; (3) “Health-related quality of life (HRQOL) in SLE”; (4) “Improving HRQOL in SLE patients”; and (5) “Focusing on psychosocial adjustment to the disease”. The investigations of the psychological outcomes of SLE are presented in Table 1 while the overview of the interventional studies is demonstrated in Table 2.

### 3.1. Anxiety, Depression and Sleep Quality

The prevalence of anxiety and depression and sleep quality in SLE was assessed in thirteen investigations in eight countries in Europe, Asia, and North America [18,19,21,22,26,27,30,34,35,38,40,42,43]. Most studies applied a cross-sectional design and used sample sizes ranging from 29 to 405 participants with an overwhelming majority (80–100%) of female participants. Studies applied a number of commonly used standardized questionnaires and tools including the Beck Depression Inventory (BDI) [19,38], the Hospital Anxiety and Depression Scale (HADS) [27,34,36,37,39,40], the Patient Health Questionnaire-9 [21], and the Hamilton Depression Rating Scale (HAMD) [18] to assess symptoms of depression and anxiety while the Pittsburgh Sleep Quality Index (K-PSQI) was used to assess sleep quality indicators [26,34,35,38]. Studies reported mixed findings as mild to severe depression was present in 15-68.7% of subjects [18,21,30]. One study, however, indicated the prevalence of depression in SLE subjects and in controls was similar [27]. According to another study, suicidal ideation could be found in as many as 14% of subjects with SLE. Suicidal ideation correlated with a younger age at diagnosis [21].

Depression was associated with anxiety, fatigue, pain intensity, symptoms of ill physical health, or medication side effects (e.g., an uptake of steroids, kidney function, or dizziness) [18,35], marital status, or ethnic background [19,21]. There were discordant findings on the relationship between depression, socioeconomic status, cognitive function, and disease activity in terms of SLEDAI scores [18,19,37,43]. A higher disease activity in terms of SLEDAI scores predicted depression [18,19,27].

An investigation on the psychological aftermath of the COVID-19 pandemic also revealed depression in SLE patients and was associated with compliance towards coronavirus prevention measures [42].

Anxiety seemed to be more common in older patients and strongly correlated with signs of the disease (e.g., alopecia, proteinuria, negative anti-P0 antibodies, poor kidney function, and fatigue) and depression [18]. Additionally, there was a positive association between anxiety and the duration of the disease (time since diagnostic disclosure) [43]. A longer duration of the disease was more anxiety-provoking only in individuals with SLE [43], but the association between the duration of the disease and anxiety in patients with JSLE could not be confirmed [21]. Furthermore, anxiety corresponded to a patient’s perception of SLE, which pertains to the patient’s perspective of and understanding of the illness (their knowledge, memories, thoughts, beliefs, and expectations about SLE) [38]. Interestingly, 19–57.4% of respondents reported anxiety whereas 57.1% complained of poor sleep [34,38]. Studies have found that poor quality of sleep in SLE patients are strongly correlated with symptoms of depression and anxiety and are predicted by older age [34,38].

### 3.2. Fatigue and Pain

Fatigue and pain in SLE were evaluated in twelve investigations carried out in eight countries in Europe, Asia, and North America [19,22,23,26,27,30,36,38,39]. The sample sizes in the respective studies ranged from 29 to 405 participants with women making up 80–100% of study participants. Most investigations were carried out by very commonly used and standardized tools, such as the Brief Pain Inventory (BPI) [22,30], the Pain Coping Questionnaire (PCQ) [30], the Pain Catastrophizing Scale (PCS) [22,30] for pain and pain coping, and the Visual Analogue Scale (VAS) [19,36,38] for pain and fatigue, while the Fatigue Severity Scale (FSS), the Vitality subscale from SF-36 [40], and the Functional Assessment Chronic Illness Therapy Fatigue (FACIT-Fatigue) [23,41] were used for fatigue. A proportion of 65–78% of respondents reported significant fatigue [30,36], and 40% of them reported clinical pain [30] and 22% pain catastrophizing [30]. SLE patients had poorer fatigue outcomes than controls [23]. Fatigue, pain intensity, and pain catastrophizing strongly predicted HRQoL and depression in individuals with SLE [26,27,36,38,39,40].

### 3.3. Health–Related Quality of Life in SLE

HRQoL in SLE was assessed in nineteen investigations in thirteen countries in Europe, Asia, and America. One study involved international respondents [19,20,22,24,25,28,29,31,32,33,34,35,36,37,40,41,42,43]. The total number of participants amounted to 6319 respondents while sample sizes ranged from 29 to 1803. The proportion of female respondents in the sample reached 80–100%. Most studies applied generic and disease-specific questionnaires, such as the SF-36 [19,23,26,27,31,32,33,35,36,37,39,40,41] and the LupusPRO [28,29,34], but one study applied qualitative methods [25]. Lupus was associated with poor HRQoL outcomes and both the JSLE and SLE patients had worse HRQoL than the controls [30,41,43]. Moreover, patients with SLE reported worse oral HRQoL despite better oral hygiene habits [20]. Cross-sectional and longitudinal studies have indicated that HRQoL is positively associated with self-management skills and regular physical activity [33]. Poor HRQoL scores were associated with lower educational status, smoking, sleep quality, fatigue, disability, organ involvement, pain catastrophizing, problematic support, and symptoms of depression and anxiety [21,24,30,31,32,33,34,35,37,39,41]. To add, lupus nephritis symptoms were strongly associated with poorer HRQoL outcomes controlling for age, ethnicity, gender, and country of residence [28]. Furthermore, males tended to have poorer scores than females [29]. There were inconsistent findings on the association between HRQoL in SLE, age, disease activity, pain severity, and treatment regimen [22,31]. In Feldman’study [25] on patients’ unmet needs for informative, emotional, and self-esteem support, participants reported feelings of loneliness and social isolation, and a need for informative support on SLE at the diagnostic disclosure.

### 3.4. Focusing on Psychosocial Adjustment to the Disease

The influence of psychological interventions for improving self-management skills and medical adherence in SLE patients was evaluated in five randomized controlled trials in China and in the U.S. [45,46,49,51,52]. The sample sizes varied from 19 to 125 while the number of respondents totaled 506. The proportion of women in each study ranged from 84–100%. The interventions addressed a plethora of aspects of psychosocial adjustment with the illness, such as: health locus of control [45], improving positive health behaviors [49,51], medical adherence [46,49,50,52], patients’ needs for social support [49], and self-efficacy [49,51]. The interventions employed digital adherence technologies [46,48,51], CBT intervention [45], stationary and social-media-based patient education [48,49,50,51], focus groups, and support groups with various meeting cadences and duration [49]. Scalzi’s study [49] found that an Internet-based educational intervention was associated with significantly enhanced medical adherence, self-efficacy, sense of agency, and empowerment scores in the experimental group compared to the controls. Xie’s [52] study captured the significant benefits of a 12 wk post-hospital transitional care intervention on positive health-behaviors, self-care, and adjustment to illness. White [51] observed insignificant improvements in the perceived lupus self-efficacy, patient activation, self-efficacy, and lower disease activity in the experimental group only. Similarly, Harry’s study [46] yielded only non-significant pre- and post-intervention differences in medical adherence in the intervention group. Nonetheless, Brown’s study 45] could not find any differences in outcome measures between two experimental groups receiving psychoeducation and cognitive behavioral treatment, and controls who did not receive treatment.

### 3.5. Improving HRQoL in SLE Patients

The efficacy of HRQoL interventions for SLE patients was evaluated in four randomized controlled trials carried out in Brazil, Greece, Serbia, and the U.S. [44,47,48,50]. The sample sizes varied from 21 to 62 while the total number of respondents reached 230. The lowest percentage of women in a single study reached 76% [50], but there were two studies involving only females [44,45]. Two studies focused on adolescent JSLE patients [45,46,47,48,49]. The publications described a home-based and stationary personalized physical exercise [44,47,50], the CBT approach [45], patient education, and app-supported mobile procedures [48,51] with a total duration of six to twenty-four weeks. Bogdanovic [44] demonstrated the benefits of exercise on HRQoL. Similarly, Keramiotou’s study [47] indicated a six month upper limb workout routine enhanced the HRQoL in the experimental group, but the differences between the intervention group and the treatment-as-usual control group could not be found. Moreover, Sieczkowska’s investigation [50] of a home-based exercise in SLE indicated the program was well-accepted by the study participants, but statistical differences in the HRQoL between the experimental and the control group could not be indicated.

## 4. Discussion

Despite a progress in understanding the pathophysiology and improving the treatments for SLE, the experience of the disease and subsequent treatment impose a large burden on patients, their families, and the society. The goal of this paper was to present empirical research on the psychological outcomes of lupus and the currently used psychological interventions for individuals with the disease.

### 4.1. Psychological Consequences of SLE

The psychological consequences of lupus were assessed by way of popular standardized methods, such as BDI II, HADS, or SF-36. In the great majority of the reviewed studies, lupus was associated with high rates of fatigue, pain, pain catastrophizing, psychological distress, anxiety, depression, and impaired sleep. SLE patients around the world reported significantly lower scores in several HRQoL domains compared to individuals with other rheumatologic diseases and healthy controls. Despite some data inconsistencies, several indicators of QoL, such as pain and fatigue, strongly correlated with mental health indicators, such as depression and anxiety. Additionally, the relationship between HRQoL and socioeconomic variables, such as family income or educational level, was non-significant or the findings were inconsistent. However, available information on the psychological responses to illness indicated that socioeconomic disparities were strongly associated with subjective HRQoL in chronically ill children and adults [53,54]. Jolly’s study [29] illuminated the sex differences in the psychological responses to lupus. They found that men were more adversely affected by the disease than women, particularly in the social support and coping domains. Women, in turn, reported poorer outcomes in, e.g., the cognitive domain and pain. This trend may also be characteristic for other rheumatologic diseases. For example, comparative studies on the psychological aftermath of rheumatoid arthritis (RA) in males and in females [55] indicate that females with RA have significantly better scores in bodily pain, but their physical and mental health are worse. In fact, emerging evidence in medical psychology suggests that male and female psychological reactions to chronic diseases differ [56,57]. For example, the investigations on Parkinson’s disease indicated the female gender was associated with lower scores both in physical and in emotional functioning while males had poorer outcomes in the cognitive domains of HRQoL [58]. These findings highlighted a need to explore the gender differences in the psychological response to a chronic illness and the importance of personalized support for chronic conditions.

### 4.2. Interventions to Improve Patients’ HRQoL and Well-Being

The current paper also analyzed the psychological interventions for lupus patients. Overall, the quality of the interventional studies assessed was high as all of them used a randomized controlled trial design to determine the outcomes with three publications on an experimental control group design. Three observations demonstrated significant post-intervention improvements in the experimental group compared with the controls. Importantly, the results of Xie’s study [52] highlighted the significance of early intervention and the role of transitional care in the process of adjusting to a chronic disease and living a happy life with lupus.

### 4.3. The Effect of Physical Fitness Interventions on HRQoL, Depression, and General Well-Being

Three studies included in the current review analyzed the effect of physical fitness interventions on HRQoL, depression, and general well-being [44,47,50]. Although only modest statistical differences between the experimental group and the controls could be detected, exercise led to improvements in the respondents’ well-being and the vital mental health indicators in the study participants. These outcomes are consistent with previous findings on the impact of structured physical activity on well-being and HRQoL in individuals with a chronic disease [59]. It may be concluded that regular low impact cardio workouts, which improve mobility and build stamina but are easy on the joints, can be recommended for SLE patients and could be combined with pharmacological treatment. Future investigations should include larger samples and the interventions should be longer to evoke the lingering beneficial effects for patients’ health.

This review has some limitations. First, only a small number of interventions presented in this review depicted the so-called “talk therapies” to elicit change and increase well-being. Nonetheless, the term “psychological intervention” has not been precisely defined yet and continues to attract attention in the literature [60,61]. In this review, “psychological intervention” pertained to treatment procedures that aim to enhance patients’ psychological well-being and various aspects of health-related quality of life. To add, the observations collected here varied in terms of sample size, experimental procedure, approaches, and outcome measures.

Second, although the methodological aspects related to experimental design were duly considered in the presented interventions, their outcomes may have resulted from a number of variables that were not considered, such as cultural, religious background, or permanent urban/rural residence. Since these factors may also impede or facilitate participation in psychological interventions, they should be carefully investigated because of stigma towards mental health services and towards patients. Consequently, the process of the implementation of psychological interventions should be investigated at the level of the healthcare provider (HCP), patient, and at the organizational/corporate level to stimulate inclusion mechanisms and patient-HCP partnership.

From a methodological point of view, future randomized controlled trials should include placebo groups as an addition to multiple experimental group designs to objectivize the interpretation of the results.

Third, although cognitive symptoms of lupus such as “brain fog” are very well-known, only very few epidemiological studies or interventions identified in the review addressed research themes related to cognitive dysfunction in the course of SLE and NPSLE [62]. Hence, continued research is needed towards developing neurorehabilitation procedures to mitigate the negative effects of the disease and enhance patients’ cognitive, social, and vocational skills. All in all, this review contributes to the existing literature on the psychological aspects of SLE as valuable adjuncts to medical care and highlights the possibilities for continued research on the topic.

## 5. Conclusions

Current studies should be revisited with the emergence of stronger evidence for the efficacy of specific psychological intervention procedures. The impact of intervention procedures may be mediated by respondents’ cultural and religious background or permanent urban/rural residence. Hence, these factors should be carefully considered while developing experimental designs.Evidence-based research is needed on effective psychological interventions for cognitive dysfunction in SLE and neurorehabilitation to ameliorate its adverse effects.

## Figures and Tables

**Figure 1 ijerph-19-16021-f001:**
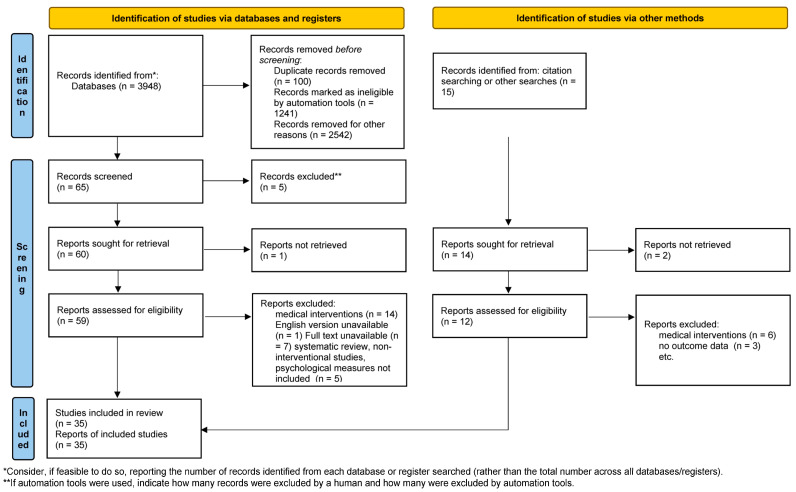
PRISMA 2020 flow diagram for new systematic reviews which included searches of databases, registers and other sources. (PRISMA 2020 statement, https://www.prisma-statement.org// (accessed on 7 October 2022).

**Table 1 ijerph-19-16021-t001:** Findings on the psychological outcomes of SLE in chronological order.

First Author, Year, and Location	Design	Sample Size (N=); Demographics and Disease Characteristics *	Aim/Domains Assessed	Assessment Tools	Pertinent Findings on Outcomes of Interest
Pettersson et al. (2012), Sweden [39]	Cross-sectional	324 pts (91% females); age range: 18–84	The most burdensome SLE symptoms and their association with demographics, disease duration, depression, anxiety, and HRQoL.	SF-36, HADS, SLE activity, and damage indices.	Fatigue, pain, and musculoskeletal distress were the most burdensome SLE symptoms. Fatigue significantly affected HRQoL.
Feldman et al. (2013), the U.S. [25]	Randomized controlled trial	29 females with SLE (age ≥ 18) from impoverished areas	Three 75 minute focus groups with a 15 min briefing session.	Focus groups on patients’ needs for informative, emotional, and self-esteem support.	Participants reported a need for informative support on SLE at the diagnostic disclosure, felt loneliness, and social isolation.
Kiani et al. (2013), the U.S. [31]	Longitudinal study	200 SLE patients (90% females) in the atorvastatin vs placebo clinical trial (Lupus Atherosclerosis Prevention Study, LAPS); average age: 44.3 ± 11.4.	To assess the association between treatment regimen (the statin vs placebo) and HRQoL at baseline, 1 year, and 2 years.	SLE activity indices, SF-36.	Significantly lower HRQoL in both (statin and placebo) groups compared to reference values. Statin use not associated with HRQoL differences. High school education predicted higher scores in physical function domain of HRQoL.
Pettersson et al. (2015), Sweden [40]	Cross-sectional	305 SLE pts and 311 matched controls	Lifestyle, fatigue, anxiety, depression.	Multidimensional Assessment of Fatigue scale (MAF), Fatigue Severity Scale (FSS), SF-36 Vitality domain, lifestyle questionnaires.	Greatest fatigue levels associated with poorest depression, anxiety, and HRQoL outcomes and least amount of exercise. Lowest fatigue associated with least symptoms of anxiety, no depression, best HRQoL scores, lowest prevalence of tobacco use, and highest reported proportion of exercising ≥ 3 times/wk.
Yilmaz-Oner et al. (2015),Turkey [43]	Comparative study	113 SLE pts and 123 matched controls	The effect of demographics, SLE duration, and activity, anxiety, depression on HRQoL.	SF-36, HADS, SLE activity scores.	SLE associated with worse HRQoL. Anxiety, depression, and being unemployed associated with worse HRQoL in SLE.
Bai et al. (2016), China [18]	Cross-sectional	176 SLE pts (86.9% women); age range: 13–52	Depression, anxiety, disease activity indices.	Hamilton Depression Rating Scale (HAMD), Hamilton Anxiety Rating Scale (HAMA), SLE activity indices.	Mild to severe depression was present in 68.7% of subjects and anxiety in 57.4% of subjects. Depression was predicted by higher disease activity. Anxiety was associated with age, alopecia, and health status, and strongly correlated with depression and disease duration.
Jones et al. (2016), the U.S. [30]	Cross-sectional, single-center	60 childhood-onset lupus (cSLE) patients (80% females); average age: 16.1 ± 2.5	Fatigue, pain, pain coping, pain catastrophizing, sleep, HRQoL, mood, anxiety, and disability.	Fatigue Scale (FS); VAS—pain, BPI, Pain Coping Questionnaire (PCQ), Pain Catastrophizing Scale (PCS), The Adolescent Sleep Wake Scale (ASWS), PedsQL, Children’s Depression Inventory I (CDI-I), the Screen for Child Anxiety Related Emotional Disorders (SCARED), The Functional Disability Inventory (FDI).	Poorer pediatric QoL scores in SLE than in controls. Significant fatigue in 65%, clinical pain in 40%, clinical anxiety in 37%, depression in 30%, and pain catastrophizing present in 22% of SLE pts. QoL predicted by pain, fatigue, and anxiety. QoL in rheumatologic disease predicted by fatigue, and pain.
Mirbagher et al. (2016), Iran [34]	Cross-sectional, single-center	77 females with SLE; average age = 36.5 ± 10.1	Sleep quality	PSQI, Hospital Anxiety and Depression Scale (HADS), and LupusQoL.	57.1% of subjects reported poor sleep, which adversely affected QoL. Poor sleep was determined by older age, anxiety, and depressive symptomatology.
Mok et al. (2016), China [35]	Cross-sectional	376 pts (95% females); average age 40.2 ± 12.9; mean disease duration 9.3 ± 7.2.	The relationship between symptoms of depression/anxiety, health-related quality of life (HRQoL), and work ability.	HADS, SF-36, SLE activity and damage indices.	15% met criteria for depression and 19% for anxiety. Depression and anxiety associated with significantly worse HRQoL controlling for demographics, SES, religion, and disease activity in the previous year.
Mazzoni et al. (2017), Italy [33]	Longitudinal, single-center	162 SLE patients	The influence of self- efficacy and dysfunctional (problematic) social support on HRQoL after 11 months.	SF-36.	Controlling for medication, HRQoL was strongly and positively affected by baseline self-management skills in SLE and adversely affected by dysfunctional (problematic) support.
Corrêa et al. (2018), Brazil [20]	Comparative study	75 SLE subjects and 78 matched controls without SLE	Oral HRQoL, disease activity and damage.	SLE activity indices and oral health impact profile (OHIP-49)	Patients with SLE reported better oral hygiene, but their oral HRQoL was worse than in controls. Dental prosthesis wearing and SLE damage predicted worse oral HRQoL outcomes.
Davis et al. (2018), the U.S. [21]	Cross-sectional, single-center	51 patients with childhood-onset SLE (cSLE) (92.2% females); age range: 7–22; mean age: 16.2 ± 3.1; mean disease length: 3.0 ± 2.9).	Association between depression and medication non-adherence	Patient Health Questionnaire-9 and Medication Adherence Self-Report Inventory.	59% screened positive for depression and 14% reported suicidal ideation and 20% reportednon-adherence. Non-adherence was more prevalent in patients with longer disease length and correlated with depression. Suicidal ideation strongly correlated with younger age at diagnostic disclosure.
Donnelly et al. (2018), the U.S. [22]	Follow-up, single-center	50 subjects with cSLE (84% females) without a history of chronic or psychiatric condition; age range: 11–20, average age: 16.2 ± 2.5	Fatigue, pain, catastrophic thinking about pain, depressive symptoms, anxiety and phobia, HRQoL and functional disability, cSLE activity and damage caused by cSLE.	Brief Pain Inventory (BPI), Pain Catastrophizing Scale (PCS), Children’s Depression Inventory Ver. I (CDI-I), Screen for Child Anxiety Related Emotional Disorders (SCARED), Functional Disability Inventory (FDI), Pediatric Quality of Life Inventory (PedsQL), generic core scales and disease-specific modules, disease activity scores.	HRQoL at follow-up was determined by fatigue and depressive symptoms at baseline, but not by age, disease activity or pain severity. The risk for poor HRQoL was associated with poorer disability, pain intensity, pain catastrophizing, and anxiety.
Elera-Fitzcarrald et al. (2018), Peru [24]	Cross-sectional, multi-center	277 pts (91.7% women); mean age at diagnosis = 41.5, median disease length = 6.5.	Disease-specific HRQoL, SLE activity and damage.	LupusQoL, SLE activity and damage indices.	Older age at diagnostic disclosure, greater disease activity and damage, and immunosuppressive therapy predicted poorer HRQoL. Better socioeconomic status (SES), disease duration, and antimalarial drug regimen predicted better HRQoL.
Figueiredo-Braga et al. (2018), Portugal [26]	Longitudinal, comparative	72 female SLE pts and 34 controls; mean age: 46.64 ± 11.49; mean disease duration: 15.11 ± 8.47 years	HRQoL, fatigue, anxiety, depression, sleep quality indices, intimate relationship satisfaction	Chalder Fatigue Scale (CFS), Fatigue Severity Scale (FSS), The Hospital Anxiety and Depression Scale (HADS), PSQI, SF-36, Couples Satisfaction Index (CSI), Relationship Assessment Scale (RAS).	Depression was manifested by 41.7% of pts. Depressive symptoms correlated with pain, anxiety, fatigue, and with alopecia and relationship satisfaction in longitudinal assessment. Fatigue severity, physical function, physical role function, and mental health best predicted depression in SLE.
Jolly et al. (2018), international [28]	Cross-sectional, multi-center	1259 pts (93% females); 43% patients with Lupus Nephritis (LN); average age: 41.7 ± 13.5.	HRQoL in LN and SLE	Lupus Patient-Reported Outcome (LupusPRO)	Presence of LN symptoms strongly associated with poorer HRQoL outcomes controlling for age, ethnicity, gender, and country of residence.
Moon et al. (2018), South Korea [37]	Cross-sectional, multicenter	152 middle aged female pts with SLE with and without fibromyalgia (FM) and 139 matched controls.	HRQoL, sleep quality, and disease activity	SF-36, EuroQol, EQ-5D, the Pittsburgh Sleep Quality Index (K-PSQI).	Worse HRQoL in SLE associated with FM. Lower educational status, SLE progression and severity, and sleep quality independently predicted HRQoL in SLE pts with FM, whereas SLE progression and severity, sleep quality, and depressive symptoms independently predicted HRQoL in non-FM subjects.
Nowicka-Sauer et al. (2018), Poland [38]	Cross-sectional	80 pts (87.5% women) mean age: 41.56.	Psychological, clinical, and sociodemographic predictors of illness perception in SLE	The Brief Illness Perception Questionnaire, State Trait Anxiety Inventory, Beck Depression Inventory (BDI), Pittsburgh Sleep Quality Index (PSQI), VAS-Pain and Fatigue	Illness perception strongly associated with state anxiety and depression variance. Middle-aged patients presented most negative perception of SLE, the poorest anxiety, depression, pain, fatigue, and sleep quality outcomes.
Piga et al. (2018), Italy [41]	Comparative study	101 SLE pts; a control group of 35 rheumatoid arthritis (RA) pts and 37 healthy controls (HCs).	The effect of musculoskeletal symptoms of SLE on disease-specific health-related quality of life (HRQoL) impairment in SLE patients	SF-36v2, FACITv4, the Heath Assessment Questionnaire (HAQ).	HRQoL in SLE subjects was worse than in controls. Erosive and deforming arthritis predicted poorer HRQoL than non-deforming non-erosive arthritis. Jacoud arthropathy, active arthritis, and fibromyalgia (FM) independently predicted poor HRQoL; fragility fractures, deformities, and active arthritis predicted disability perception in SLE.
Jolly et al. (2019), the U.S. [29]	Cross-sectional, multi-center	1803 pts (93.3% women); mean age: 40.8 ± 13.1	Comparison of the quality of life in lupus outcomes between males and females.	LupusPRO, disease activity and damage indices	Males were more adversely affected by the disease than females with significantly worse results in perceived social support and coping domains than females. Females were more affected in cognition, procreation, physical health, pain and vitality domains.
Elefante et al. (2020), Italy [23]	Cross-sectional	158 SLE pts (91.9% females); mean age: 44.9 ± 13.2, median disease duration: 13 years and 65 matched healthy controls (HC)	HRQoL, pain, fatigue, the SLE effects on cognition, emotional health status, body image and life plans.	SF-36, The Functional Assessment Chronic Illness Therapy Fatigue (FACIT-Fatigue), SLE activity scores.	SLE subjects had poorer fatigue results than HCs. Fatigue outcomes associated with fibromyalgia and self-reported SLE activity.
Louthrenoo et al. (2020), Thailand [32]	Longitudinal, comparative study	337 patients (96% females); mean age: 37; median age at diagnosis: 26; median disease duration: 7.	Longitudinal effect of disease activity on HRQoL	SLE Quality of Life (SLEQOL) and SF-36	HRQoL and quality of life in lupus associated with the extent of organ involvement and disease activity.
Tee et al. (2020), Philippines [42]	Cross-sectional	405 (79.1%) SLE patients and 107 (20.9%) Rheumatoid Arthritis (RA) pts (95.5% females); age range: 22–50	Recent health service utilization, pandemic-related health literacy, adherence and attitude towards prevention and control measures for COVID-19, comorbidities, availability of medication and recently experienced disease symptoms, depression, anxiety, pandemic-related distress.	Depression, Anxiety and Stress Scale (DASS-21), the Impact of Event Scale-Revised (IES-R).	SLE associated with significantly higher overall and COVID-related distress, anxiety, and depression scores. Impaired mental health status was predicted by, e.g., healthcare occupation, comorbidities, hypertension, pain, breathing difficulty, and dizziness. Information satisfaction on pandemic predicted stress, depression, and psychological impact of COVID-19. Adherence to COVID-19 prevention measures correlated with lower stress and depression.
Figueiredo-Braga et al. (2021), Portugal [27]	Comparative study	29 JSLE pts (90% females); mean age: 23.1 ± 5.4; a control group of 73 adult-onset SLE patients, 22 HCs and 32 depressed pts (100% women)	The association between JSLE clinical manifestations and depression and anxiety in the course of the disease.	Fatigue Severity Scale (FSS), HADS, SF-36.	JSLE subjects had low depression scores (similar to healthy controls), but poor anxiety outcomes. Fatigue and kidney function markedly correlated with depressive symptoms in JSLE. Symptoms of depression/anxiety associated with received treatments (e.g., steroids) and with damage, but not with disease activity in SLE.
Monahan et al. (2021), The Netherlands [36]	Observational	348 SLE pts including 21% pts with NPSLE (87%, females); mean age: 43 ± 14.	Anxiety, depression, and fatigue	HADS, Vitality (VT, from SF-36), the Multidimensional Fatigue Inventory (MFI), VAS-Fatigue.	Fatigue was present in 78% of pts. Fatigue significantly associated with anxiety and depression symptoms, but not with disease activity.
Chalhoub et al. (2022), the U.S. [19]	Cross-sectional	99 SLE pts (93% women); mean age: 46.4 ± 12.1; mean disease length: 10.6 ± 6.8, mild to moderate disease activity.	Pain, cognitive function, the prevalence of moderate/severe symptoms of depression (MSD), HRQoL mental domains.	Automated Neuropsychological Assessment Metrics (ANAM), Beck Depression Inventory II (BDI II), SF-36, Visual Analogue Scale (VAS)–Pain, SLE activity indices.	MSD was influenced by family income, but not by educational status, ethnic background, marital status, or high-dose medication. MSD was independently predicted by disease activity and pain. BDI II outcomes strongly correlated with cognitive function, disease activity, and pain intensity, but not with educational level, family income, or cognitive function. Age, BDI II results, and pain significantly correlated with the physical domain of HRQoL.

* Chronological data such as age and duration in years unless otherwise specified.

**Table 2 ijerph-19-16021-t002:** An overview of the interventional studies.

First Author, Year, and Location	Design	Sample Size (N=); Demographics and Disease Characteristics *	Type and Schedule of the Intervention	Assessment Tools	Findings
Interventions focused on improving HRQoL in SLE patients					
Physical exercise interventions					
Bogdanovic et al. (2015), Serbia [44]	Randomized controlled trial	60 female SLE pts; mean age: 43.4 ± 12.8	6 weeks intervention of 15 min bicycle and ergometer or 30 min isotonic exercises, 3 days a week	(FSS), SF-36, BDI	Significant improvements in depression and QoL in both groups.
Keramiotou et al. (2020), Greece [47]	Randomized controlled trial	62 SLE pts (93.5% females); mean age: 43.3; with arthralgias and upper limb disabilities	30 min daily personalized upper limb exercise for 24 weeks	The DASH questionnaire to assess symptoms and function of the entire upper extremity, Health Assessment Questionnaire (HAQ)	Significantly improved QoL, fatigue, pain scores, and limb function in the intervention group. No significant intervention/control group differences.
Sieczkowska et al. (2022), Brazil [50]	Randomized controlled trial	21 patients with JSLE and Juvenile Idiopathic Arthritis (JIA) (76% females); age range: 10–19; mean age: 15.8 ± 2.25 years; and 30 controls.	12 wk home-based indoor tri-weekly workout intervention with an instructor.	Strengths and Difficulties Questionnaire (SDQ), PedsQL 4.0 Generic Core Scale, PSQI,39	Adherence to the exercise protocol reached 76.7% for JSLE vs. 44.8% in the JIA. No significant intervention/control group differences in HRQoL, well-being, or sleep quality.
		Digital e-health interventions			
Khan et al. (2020), the U.S. [48]	Randomized controlled trial	34 SLE pts (96.5% females); mean age: 42	16 wk lifestyle-smartphone app and e-health coaching; therapeutic intervention	FACIT-F, BPI-Short Form, and Lupus Quality of Life (LupusQoL).	Significant intervention/control group differences in all domains.
	Remotely-delivered interventions focusing on psychosocial adjustment to SLE				
Brown et al. (2012), the U.S. [45]	Randomized controlled trial	53 female JSLE pts; age range: 12–18); two intervention arms and care-as-usual controls	6 wk cognitive- behavioral therapy (CBT) intervention on mood and activity monitoring, relaxation, coping, problem-solving, time management. 6 wk education arm—impact of SLE on daily functioning, healthy lifestyle.	The McGill Pain Questionnaire—Short Form (SF-MPQ)18, The Behavior Assessment System for Children (BASC), The Positive and Negative Affect Schedule-Extended Version (PANAS-X), The Self-Perception Profile for Adolescents (SPPA), The Multidimensional Health Locus of Control Scales (MHLC), The PedsQL Rheumatology	No significant differences between the CBT intervention group and controls.
Scalzi et al. (2018), the U.S. [49]	Randomized controlled trial	27 JSLE pts (96% females); mean age: 18.1	Eight Internet-based social media educational intervention sessions and peer support group focused on self-management skills.	Medication possession ratio (MPR), MASRI, Perceived Severity of Stress Questionnaire (PSQ), the Children’s Arthritis Self-Efficacy scale (CASE), the Simple Measure of the Impact of Lupus Erythematosus in Youngsters (SMILEY) index	Significantly increased medical adherence, self-efficacy, sense of agency, sense of community, and empowerment scores compared to controls.
Xie et al. (2018), China [52]	Single-center, single-blind, randomized controlled trial	125 SLE pts (88.8%, females) mean age: 37.1	12 wk post-hospital transitional care phone intervention on adaptation to illness, health behaviors, and QoL.	The Exercise of Self-Care Agency Scale, SF-36.	Significantly better QoL and self-care, lower hospital readmission rate than controls.
Harry et al. (2020), the U.S. [46]	Single-blind randomized controlled trial	19 cSLE pts (84% females with similar severity of pain and fatigue); mean age: 20.5	Intervention based on adherence-tracking pillbox and digital reminder to facilitate adherence.	Barriers to Adherence Tool (BAT), PROMIS^®^ Pain Numeric Rating Scale v.1.0, PROMIS Fatigue Short Form v1.0 (Fatigue 4a), Feasibility and acceptability questionnaire, Medication adherence self-report inventory (MASRI), a SimpleMed+ pillbox and automated digital reminders for objective adherence	Non-significant increase in adherence in the intervention group. Forgetting identified as the main obstacle to adherence. Acceptance rate of the intervention was 50%.
White et al. (2021), the U.S. [51]	Randomized controlled trial	30 SLE patients; age range: 18–65	12 wk self-management intervention based on navigator-led weekly phone sessions to improve adherence, adjustment to the disease, and well-being.	Chew Health Literacy Scale, Lupus Self-Efficacy Scale, the patient activation measure (PAM), Systemic Lupus Activity Questionnaire (SLAQ), PHQ-9, the 7-item General Anxiety Disorder scale, LupusQoL, SF-36 FACIT-F, satisfaction/ dissatisfaction with healthcare.	Significantly improved perceived lupus self-efficacy, non-significantly improved patient activation, self-efficacy, and lower disease activity in the experimental group. No significant intergroup differences in health literacy.

* Chronological data such as age and duration are provided in years unless otherwise specified.

## Data Availability

Not applicable.

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
