# Peer review of "Psychological Implications to the Therapy of Systemic Lupus Erythematosus"

_ijerph, 2022, doi:10.3390/ijerph192316021_

Round 1
Reviewer 1 Report
Authors have comprehensively selected and commented on papers regarding the psychological aspects of SLE. However, there are some ambiguities in the paper. Please consider revising.
In L147 Authors says “Anxiety strongly correlated with depression and was associated with fatigue, age, perception of SLE, SLE symptoms (e.g. alopecia), and time since diagnosis“.
However,
What exactly is “perception of SLE”?
What age groups are most anxious?
Is a longer period of time since diagnosis more anxiety-provoking? Is a shorter period of time more anxiety-provoking?
Besides Alopecia, what other symptoms are related to anxiety?
These are information that clinicians would like to know, so it would be better to include them specifically in the text.
Author Response
Poznan, Nov. 20, 2022
Dear Editor and reviewers,
Thank you very much for giving us the opportunity to submit a revised draft of the manuscript Psychological implications to the therapy of systemic lupus erythematosus (ID 1986299) for publication in the International Journal of Environmental Research and Public Health. We sincerely appreciate your precious time in reviewing our paper and all valuable comments and suggestions, which made it possible for us to improve the quality of the article. In the current version of our manuscript we enabled the "track changes" option so that all the corrections made in the current version are highlighted within the manuscript. The following responses have been prepared to address all of the referees’ comments in a point –by-point fashion.
Referee: 1
Comments to the Author
In L147 Authors says “Anxiety strongly correlated with depression and was associated with fatigue, age, perception of SLE, SLE symptoms (e.g. alopecia), and time since diagnosis“.
However,
What exactly is “perception of SLE”?
Thank you very much for this comment. This information was missing indeed. Perception of systemic lupus erythematosus pertains to patient's perspective of and understanding of the illness (their knowledge, memories, thoughts, beliefs and expectations about SLE). Perception of an illness may affect patient's psychosocial well-being and overall behavior including health-related behaviors (e.g., medication adherence and treatment compliance). We have added the description of "perception of SLE" within a manuscript.
What age groups are most anxious?
We added the information that anxiety was positively associated with older age.
Is a longer period of time since diagnosis more anxiety-provoking? Is a shorter period of time more anxiety-provoking?
Studies indicated longer duration of the disease was more anxiety-provoking only in individuals with SLE but not in JSLE. We added the information in the text.
Besides Alopecia, what other symptoms are related to anxiety?
Thank you for pointing this out. Anxiety was associated with signs of the disease such as alopecia, proteinuria, negative anti-P0 antibodies, poor kidney function and depression. Disease activity in terms of SLEDAI scores predicted depression in adults with SLE but not in younger patients.
These are information that clinicians would like to know, so it would be better to include them specifically in the text.
Thank you very much again for your valuable comments
Best regards
The corresponding author

Reviewer 2 Report
This paper presents a comprehensive review of the available literature surrounding psychiatric disease in SLE - this remains a complicated area in the treatment of SLE and this paper summarizes well our present understanding in this respect. This paper additionally looked at both observational and interventional studies, separating these two groups in the summary of the available studies, which helps to provide a more nuanced perspective on the available literature.
Overall I feel that this review addresses an area of SLE research that continues to be a huge area of need, and so this serves as an excellent overview of the available literature. The search for relevant articles appears to be comprehensive, and the article has also organized the results into several key themes which helps to understand the volume of information that is available.
Detailed comments:
- Some of the statements in the introduction should be supported by appropriate references. Specifically, the statements about the epidemiology and potential reasons for the variable course of the disease (lines 33-35) should have references.
- On line 39-40, you describe that neuropsychiatric disease indicates the highest disease activity. I think it would be more accurate to say that "neuropsychiatric lupus has one of the heaviest weights in the measurement of SLE disease activity" - it depends on the instrument, but for example, vasculitis is weighted just as heavily as neuropsychiatric manifestations on the SLEDAI-2K so it would be inaccurate to say that neuropsychiatric disease is the highest.
- On line 118, "HRQOL" is used without this having been used in non-abbreviated form previously - would suggest writing it out in full before using the abbreviation. Similarly, there are a few spots where "systemic lupus erythematosus" is used in full despite having used the abbreviation "SLE" previously - could replace all of these with the abbreviation to save words.
- I would suggest trying to clean up Table 1 for readability. Specifically, it would be nice if: a) studies were organized chronologically or by study design (I think it's by first author right now?) as I think it might lead to a more coherent story either in terms of how the discussion around neuropsychiatric outcomes has changed over time, or whether there are shared findings amongst different study types, and b) the information about demographics/disease characteristics was presented in the same way for each study (there are many different variations on the way it is expressed in the table). Will also add that the capitalization for the study design field is inconsistent so would be good to unify this.
- For Table 2, I think similarly the presentation could be improved if studies in the same theme were grouped together (e.g. all the exercise interventions were put together). This would help in drawing out relevant key findings as well (see my comment about the discussion below).
- In Table 1, the row with the study by Correa et al. - there is something wrong with the sentence in the "Pertinent findings" column as it does not make sense - I think there might be an extra "worse"?
- In Table 1, the row with the study by Elera-Fitzcarrald et al. - "SES" is used in the findings column but has not been used in full previously.
- Line 136: in the several sentences in this area, there are a number of statements referring to studies but there are no attached references. The appropriate study should be referred to with each statement. This is true for statements made in other sections about the relevant studies as well.
- Line 242: you make reference to emerging data showing that there are gender differences in terms of reacting to chronic disease - there needs to be a referenced study here however.
- Line 282: you introduce NPSLE here for the first time but I think you could have inserted this abbreviation far ahead in the beginning of your paper and made use of this abbreviation throughout.
- I like how the discussion boils down the findings into some key messages (e.g. the relationship between HRQOL and socioeconomic variables). One of the key things we always think about in SLE is whether there is any relationship to disease activity (since this is what we target in trying to treat SLE) - would be nice if you could add a statement about any relationship to disease activity that you saw.
- Lines 246-248: you bring forward a study about RA, I presume to highlight your point about gender differences. It would be good to make this a bit more explicit as at first read I could not quite understand why RA was being invoked all of a sudden.
- In the discussion, it would be nice if you could talk about some key themes in the interventional studies too. For example, the interventions could be split into exercise programs vs self-management/CBT interventions - any clear messages arising from this?
Author Response
Poznan, Nov. 20, 2022
Dear Editor and reviewers,
Thank you very much for giving us the opportunity to submit a revised draft of the manuscript Psychological implications to the therapy of systemic lupus erythematosus. (ID 1986299) for publication in the International Journal of Environmental Research and Public Health. We sincerely appreciate your precious time in reviewing our paper and all valuable comments and suggestions, which made it possible for us to improve the quality of the article. In the current version of our manuscript we enabled the "track changes" option so that all the corrections made in the current version are highlighted within the manuscript. The following responses have been prepared to address all of the referees’ comments in a point –by-point fashion.
Referee: 2
Comments to the Author
This paper presents a comprehensive review of the available literature surrounding psychiatric disease in SLE - this remains a complicated area in the treatment of SLE and this paper summarizes well our present understanding in this respect. This paper additionally looked at both observational and interventional studies, separating these two groups in the summary of the available studies, which helps to provide a more nuanced perspective on the available literature. Overall I feel that this review addresses an area of SLE research that continues to be a huge area of need, and so this serves as an excellent overview of the available literature. The search for relevant articles appears to be comprehensive, and the article has also organized the results into several key themes which helps to understand the volume of information that is available.
Thank you very much for these comments.
Detailed comments:
- Some of the statements in the introduction should be supported by appropriate references. Specifically, the statements about the epidemiology and potential reasons for the variable course of the disease (lines 33-35) should have references.
Thank you very much for this valuable comment. We have added the corresponding references.
- On line 39-40, you describe that neuropsychiatric disease indicates the highest disease activity. I think it would be more accurate to say that "neuropsychiatric lupus has one of the heaviest weights in the measurement of SLE disease activity" - it depends on the instrument, but for example, vasculitis is weighted just as heavily as neuropsychiatric manifestations on the SLEDAI-2K so it would be inaccurate to say that neuropsychiatric disease is the highest.
Thank you very much for this remark. We have appropriately modified the text in accordance with your suggestions.
- On line 118, "HRQOL" is used without this having been used in non-abbreviated form previously - would suggest writing it out in full before using the abbreviation. Similarly, there are a few spots where "systemic lupus erythematosus" is used in full despite having used the abbreviation "SLE" previously - could replace all of these with the abbreviation to save words.
Thank you very much. We have corrected the text accordingly.
- I would suggest trying to clean up Table 1 for readability. Specifically, it would be nice
if: a) studies were organized chronologically or by study design (I think it's by first author right now?) as I think it might lead to a more coherent story either in terms of how the discussion around neuropsychiatric outcomes has changed over time, or whether there are shared findings amongst different study types, and
Table 1 was reorganized so that the studies are now presented in a chronological order.
- b) the information about demographics/disease characteristics was presented in the same way for each study (there are many different variations on the way it is expressed in the table).
We have modified these data so that now they are provided in a uniform way.
Will also add that the capitalization for the study design field is inconsistent so would be good to unify this.
We have corrected this error, thank you.
- For Table 2, I think similarly the presentation could be improved if studies in the same theme were grouped together (e.g. all the exercise interventions were put together). This would help in drawing out relevant key findings as well (see my comment about the discussion below).
Thank you, we have reorganized Table 2 and now the interventional studies in the same theme are grouped together.
- In Table 1, the row with the study by Correa et al. - there is something wrong with the sentence in the "Pertinent findings" column as it does not make sense - I think there might be an extra "worse"?
Thank you very much for this comment. One word was redundant indeed. We have corrected this error.
- In Table 1, the row with the study by Elera-Fitzcarrald et al. - "SES" is used in the findings column but has not been used in full previously.
The acronym SES pertains to socio-economic status. We provided the expansion of the acronym in the table.
- Line 136: in the several sentences in this area, there are a number of statements referring to studies but there are no attached references. The appropriate study should be referred to with each statement. This is true for statements made in other sections about the relevant studies as well.
We have added the references in the appropriate section of the manuscript.
- Line 242: you make reference to emerging data showing that there are gender differences in terms of reacting to chronic disease - there needs to be a referenced study here however.
We have added the appropriate references:
- Yu, T., Enkh-Amgalan, N., Zorigt, G., Hsu, Y.J., Chen, H.J., Yang, H.Y. Gender differences and burden of chronic conditions: impact on quality of life among the elderly in Taiwan. Aging Clin Exp Res. 2019, 31, 1625-1633. doi: 10.1007/s40520-018-1099-2. Epub 2019 Jan 2. PMID: 30604210.
- Gemmell, L.A., Terhorst, L., Jhamb, M., Unruh, M., Myaskovsky, L., Kester, L., Steel, J.L. Gender and Racial Differences in Stress, Coping, and Health-Related Quality of Life in Chronic Kidney Disease. J Pain Symptom Manage. 2016, 52(6), 806-812. doi: 10.1016/j.jpainsymman.2016.05.029. Epub 2016 Sep 30. PMID: 27697565; PMCID: PMC5156935.
- Line 282: you introduce NPSLE here for the first time but I think you could have inserted this abbreviation far ahead in the beginning of your paper and made use of this abbreviation throughout.
Thank you very much for this comment. we have corrected this.
- I like how the discussion boils down the findings into some key messages (e.g. the relationship between HRQOL and socioeconomic variables).
One of the key things we always think about in SLE is whether there is any relationship to disease activity (since this is what we target in trying to treat SLE) - would be nice if you could add a statement about any relationship to disease activity that you saw.
We added the information that disease activity in terms of SLEDAI scores predicted depression in individuals with SLE.
- Lines 246-248: you bring forward a study about RA, I presume to highlight your point about gender differences. It would be good to make this a bit more explicit as at first read I could not quite understand why RA was being invoked all of a sudden.
The example of RA was used to indicate the impact of sex on mental health in a chronic disease.
- In the discussion, it would be nice if you could talk about some key themes in the interventional studies too. For example, the interventions could be split into exercise programs vs self-management/CBT interventions - any clear messages arising from this?
We have added topic headings and added a short paragraph on interventional studies within the discussion section.
Thank you again
Best regards
the corresponding author